# Medical Studies during the COVID-19 Pandemic: The Impact of Digital Learning on Medical Students’ Burnout and Mental Health

**DOI:** 10.3390/ijerph18010349

**Published:** 2021-01-05

**Authors:** Panagiotis Zis, Artemios Artemiadis, Panagiotis Bargiotas, Antonios Nteveros, Georgios M. Hadjigeorgiou

**Affiliations:** Medical School, University of Cyprus, 2029 Nicosia, Cyprus; artemiadis.artemios@ucy.ac.cy (A.A.); bargiotas.panagiotis@ucy.ac.cy (P.B.); anteve01@ucy.ac.cy (A.N.); hadjigeorgiou.georgios@ucy.ac.cy (G.M.H.)

**Keywords:** burnout, pandemic, COVID-19, medical studies, cynicism

## Abstract

Objectives: The aim of this ecological study was to investigate what the impact of digital learning due to the COVID-19 pandemic was on the burnout and overall mental health (MH) of medical students. Background: During the unprecedented era of the COVID-19 pandemic, the majority of countries worldwide adopted very strong measures. Universities closed their doors, and education continued through digital learning lectures. Methods: An anonymous questionnaire was administered to all 189 eligible candidates before and during the COVID-19 pandemic. Mental health was assessed via the MH domain of the 36-item Short Form Health Survey (SF-36) and burnout with the Maslach Burnout Inventory—Student Survey (MBI-SS). Results: The overall response rate was 81.5%. The overall burnout prevalence did not differ significantly between the two periods (pre-COVID-19 18.1% vs. COVID-19 18.2%). However, the burnout prevalence dropped significantly in year 4 (pre-COVID-19 40.7% vs. COVID-19 16.7%, *p* = 0.011), whereas it increased significantly in year 6 (pre-COVID-19 27.6% vs. COVID-19 50%, *p* = 0.01). When looking at each MBI-SS dimension separately, we found that emotional exhaustion decreased significantly in year 4 but increased in year 6, and cynicism increased in all years. The overall MH deteriorated significantly between the two periods (pre-COVID-19 58.8 ± 21.6 vs. COVID-19 48.3 ± 23, *p* < 0.001). Conclusions: Digital learning in medical studies carries significant risks. Not only does the MH deteriorate, but cynicism levels also increase. Emotional exhaustion was found to increase particularly in final year students, who struggle with the lack of clinical experience just before they start working as qualified junior doctors.

## 1. Background

Burnout is a prolonged response to chronic emotional and interpersonal stressors in the working environment. Burnout is defined by the three key dimensions of emotional exhaustion, depersonalization (often referred to as cynicism), and increased feelings of inefficacy, often referred to as reduced feelings of personal accomplishment [1]. Emotional exhaustion refers to feelings of being overextended and depleted of one’s emotional resources. Depersonalization or cynicism is characterized by a negative, cynical, and detached response to other people, including colleagues, patients, or clients. A reduction in personal accomplishment occurs when a person feels less competent in their work.

Healthcare workers, including nurses [1], residents [2], and specialist doctors [3], are very vulnerable to burnout and its consequences, namely, poor work engagement [4] and poor wellbeing [5]. Burnout during medical studies is a phenomenon that is increasingly gaining attention in education research. Burnout plays a significant role in the overall wellbeing of medical students and has severe implications for the continuation of burnout during the residency period and even beyond [6]. Previous research has shown a consistent pattern of increasing burnout after the initiation of clinical exercise where students confront patients, disease, and death.

Medical students are at high risk of depression and suicidal ideation [7]. Current literature supports a strong link between burnout in medical students and increased suicidality. Cross-sectional data from seven medical schools showed that students experiencing burnout are up to 3 times more likely to have considered suicide in the past [8]. Burnout severity is also highly associated with suicidal ideation, and this association persists after having adjusted for depression within the same population. Fortunately, students who have previously reported burnout tend to recover from this state and its associated increase in suicidal ideation [8].

During the unprecedented era of the COVID-19 pandemic, the majority of countries worldwide have adopted very strong measures. Quarantine has been used for centuries to contain the spread of infection by isolating those who have (or may have) been infected. Universities closed their doors, and education continued through digital learning lectures. Lockdown, quarantine measures, and social distancing have already had detrimental effects on the mental health of people as symptoms of depression, anxiety, and stress have dramatically increased [9].

The aim of this ecological study was to investigate what the impact was of digital learning, which was implemented because of the COVID-19 pandemic, on the burnout and overall mental health of medical students at the University of Cyprus.

## 2. Methods

### Study Design

This is an ecological study that took place in the Medical School of the University of Cyprus. The Medical School Undergraduate Program runs for 6 years (three preclinical and three clinical years). A total of 189 medical students who are enrolled in the program were invited to participate to the study. Originally, the study was designed as a cross-sectional study before the COVID-19 pandemic, and thus, we had a baseline assessment (pre-COVID-19) which took place in January 2020 [10]. An anonymous questionnaire was distributed in the form of a hard copy to all students during the last week of January 2020. Participants were asked to return the completed questionnaires in a sealed envelope, which they placed in a non-transparent empty box, in order to ensure the anonymity of the questionnaire. All candidates were reminded to complete the questionnaire one week after the initial distribution.

As the pandemic emerged in February 2020 and a subsequent lockdown and digital learning took place in March 2020 (including no clinical teaching for the clinical years 4–6), we performed a follow-up assessment in May 2020 (COVID-19). The follow-up assessments were the same as in the baseline and are detailed below. The follow-up assessments were collected by an anonymous electronical questionnaire that was disseminated to students via email.

A total of 154 out of the 189 medical students (18 first year, 28 s year, 32 third year, 24 fourth year, 26 fifth year, and 26 sixth year) participated in both assessments (pre- and post-COVID-19), giving an overall response rate of 81.5%.

Patients or the public were not involved in the design, or conduct, or reporting, or dissemination plans of our research. The study was approved by the Cyprus National Bioethics Committee. All participants gave written informed consent in order to participate.

## 3. Assessments

Baseline *demographic characteristics* included age, sex, and marital status. Academic characteristics included year of studies.

*Mental health (MH)* was evaluated by the MH domain of the 36-item Short Form Health Survey (SF-36) [11]. Students answered the five items concerning their emotional wellbeing during the last four weeks in a 6-point Likert-type scale, ranging from 1 (all of the time) to 6 (never). Two items (‘‘Have you been a happy person?’’ and “Have you felt calm and peaceful?”) were reverse scored to ensure that a higher item value indicated better mental health. Then scores were transformed into a 0 to 100-point scale as follows; 1 = 100, 2 = 80, 3 = 60, 4 = 40, 5 = 20, and 6 = 0. The sum was divided by 5 in order to be averaged. Higher scores denoted better MH. The instrument showed good reliability for this study (Cronbach’s alpha = 0.88).

*Burnout*: The Maslach Burnout Inventory—Student Survey (MBI-SS) was used to evaluate burnout among medical students [12]. Licenses to use MBI-SS were purchased via Mind Garden Inc. MBI-SS is a 16-item tool, with each item being rated on a 7-point Likert-type scale, ranging from 0 (never) to 6 (everyday). These items produce three subscales: exhaustion (EX), cynicism (CY) and efficacy (EF). As suggested by Schutte et al., one particular CY item (“When I’m in class or I’m studying I don’t want to be bothered”) was removed because it was shown to be ambivalent and, thus, unsound [13]. The construct and concurrent (i.e., criterion-related) validity and reliability of this instrument were verified. The instrument showed good to excellent reliability for this study (Cronbach’s alpha: EX = 0.92, CY = 0.87, EF = 0.84).

It has been previously suggested that the most effective way of diagnosing burnout involves using a system of high scores on both EE and CY, or a high score on EX combined with a low score on EF [14]. As per our previous work on burnout [15], distribution of each subscale score of this study population was divided into quartiles, and high scores meant scoring in the 75th percentile or higher, whereas low scores meant scoring in the 25th percentile or lower. Thus, a high score on both MBI-SS-EX (21 for this study) and MBI-SS-CY (5 for this study), or a high score on MBI-SS-EX combined with a low score on MBI-SS-EF (22 for this study) were used to distinguish “burned out” from “non-burned out” students.

## 4. Statistical Analyses

A database was developed using the Statistical Package for Social Science version 21.0 (Armonk, NY, USA: IBM Corp.). Frequencies (%) were used to present data for burnout prevalence. Univariate analysis tests were performed to address the study’s aims. We applied binomial tests to compare burnout prevalence between the different COVID-19 periods, since medical students between each COVID-19 group were not completely independent nor matched, as this was an ecological study. Within-COVID-19-group comparisons for burnout prevalence across the medical years of education were made using the chi-square test. Adjusted standardized residuals served to ascertain significant deviations from the expected frequencies. One sample Wilcoxon signed rank test was applied to compare EX, CY, EF, and MH between COVID-19 groups, due to the violation of the normality assumption. The level of significance was 0.05. The corrected *p*-value of 0.02 for six comparisons was used for all COVID-19 group comparisons across the medical years of education, according to the Benjamini–Yekutieli method.

## 5. Results

The mean age of the 154 participants (69.5% females) was 22.6 ± 4.1 years (ranging from 18 to 52 years). The overall burnout prevalence among medical students did not differ significantly between the two periods (pre-COVID-19 18.1% vs. COVID-19 18.2%). During lockdown, the highest prevalence of burnout was noted in the last year of medical studies, whereas during the pre-COVID-19 period, the highest prevalence of burnout was seen in the fourth year of studies, which is the year when clinical training begins. The burnout prevalence differed significantly in these two training years when comparing the pre-COVID-19 and COVID-19 measurements.

Specifically, burnout prevalence dropped significantly in year 4 medical students from 40.7% during the pre-COVID-19 period to 16.7% during the COVID-19 period (*p* = 0.011). In addition, we found a statistically significant increase of burnout from 27.6% during the pre-COVID-19 period to 50% during the COVID-19 period (*p* = 0.010).

Table 1 summarizes the burnout prevalence across the academic years in the different COVID-19 periods.

When looking at each MBI-SS dimension separately, we found various significant differences between pre-COVID-19 and COVID-19 periods, within each academic year. In particular, the exhaustion subscale was found to be significantly decreased during the COVID-19 period compared to the pre-COVID-19 period in students at the 4th academic year (*p* = 0.002). On the other hand, exhaustion significantly increased in students at the 6th academic year during the COVID-19 period compared to the pre-COVID-19 period (*p* = 0.004). Considering the similar direction of the significant changes described above for burnout prevalence, it can be deduced that these differences could mainly be ascribed to the emotional exhaustion changes.

With regard to cynicism, students at the 1st, 2nd, 3rd, 5th, and 6th academic year reported significantly increased cynicism scores during the COVID-19 period versus the pre-COVID-19 period. Interestingly, the efficacy scores did not differ significantly between the two COVID periods, with the only exception of the 4th academic year wherein students had lower efficacy scores during the COVID-19 period than during the pre-COVID-19 period.

Overall MH deteriorated significantly between the two periods (pre-COVID-19 58.8 ± 21.6 vs. COVID-19 48.3 ± 23, *p* < 0.001). When looking into the academic years separately, MH deteriorated in all years and reached statistical significance in academic years 1 (*p* = 0.001), 3 (*p* = 0.008), and 6 (*p* = 0.001). Figure 1 illustrates the changes observed for the exhaustion, cynicism, and efficacy subscales and the mental health scores between the two COVID-19 periods within each academic year.

## 6. Discussion

The novelty of our study is that we examined what impact digital learning had in MH and burnout during the medical studies. We were able to perform this study as we had a baseline assessment only a few weeks before the announcement of the strict measures due to the COVID-19 pandemic, and we managed to re-assess students during the COVID-19 pandemic while the university still was offering only live electronic lectures. Our overall response rate was high (81.4%), which strengthens our results.

In total, burnout was found to affect almost one in five medical students, with higher levels in the clinical years. Such higher prevalence of burnout in the year when clinical training begins has been observed in numerous studies of burnout [16,17,18,19,20,21,22].

A first finding of our study is that the distribution of burned-out students across the academic years has changed. During lockdown, the highest prevalence of burnout was noted in the last year of medical studies, whereas during the pre-COVID-19, period the highest prevalence of burnout was seen in the fourth year of studies, which is the year when clinical training begins in our Medical School. Fourth-year medical students that were burnout out in the lockdown period were significantly fewer than in the pre-COVID-19 period. This phenomenon can be explained by the fact that during the fourth year, medical students start their clinical training, during which they spend most of their time within hospitals, facing patients and their family members who often suffer, and coming across terminal diseases, or even death. Specifically, direct contact with patients normally occupies two thirds of students’ daily workload during the clinical academic years (4th to 6th). When this contact stopped, as a result of the transition to digital learning, we observed a decrease of burnout in those students.

By contrast, in year 6, the opposite happened. Sixth-year medical students that were burnout out in the lockdown period were significantly more than in the pre-COVID-19 period, despite the cutdown on their clinical training. A possible explanation for this is that those students are one step before becoming doctors. This phenomenon can be explained by the fact that the sixth year is the last year of medical studies, meaning that doctors-to-be are just one step before starting to work as residents in the hospitals. Doctors, particularly at the first stages of their careers, suffer from increased stress due to the responsibilities they have to colleagues and to patients. Clinical experience often reduces such stressors. The fact that the clinical training of sixth-year doctors stopped and went virtual had a negative effect on their psychology and confidence. The uncertainty of subsequent years as professionals along with the lack of clinical experience can be very burdensome.

When looking at each dimension of burnout separately, we found that in all years, CY levels increased. This dimension of MBI-SS assesses the views of students about their studies [10]. Increased cynicism means that students doubt the significance and usefulness of their studies and become less interested or enthusiastic. A prerequisite for becoming a medical doctor is getting clinical experience, which cannot happen through electronic lectures or videos.

Moreover, different patterns were observed in the exhaustion dimension. The emotional exhaustion reported by 4th-year medical students decreased significantly, whereas it increased significantly in 6th-year students. This accounted for the change in the overall burnout rate when looking at each academic year separately; the rate dropped in the 4th year, while it increased dramatically in the final year.

Mental health also deteriorated in our study population during lockdown. This is in keeping with other studies, which demonstrates a significant increase in depression levels as the pandemic was progressing [23]. This finding, however, should be interpreted with caution given the fact that there are confounding factors, other than digital learning alone, which may have contributed to the overall deterioration of mental health. In particular, social isolation which took place during lockdown may have also had a significant effect in overall mental health. The effect of social isolation on mental health is evident through numerous studies and has already raised significant concerns [24,25,26]. It is argued that negative mental health impacts do not simply stop but continue following the lockdown period [27]. Social isolation associated with quarantine can be the catalyst for many mental health disorders, even in people who were previously mentally healthy. Such mental health sequalae include but are not limited to acute stress disorders, irritability, insomnia, emotional distress, and mood disorders, including depressive symptoms, fear and panic, anxiety, and stress. The length of lockdown plays a significant role in these phenomena [27].

Our findings should be interpreted with caution given the limitations of our study. Firstly, this is an ecological study, and therefore, we cannot make etiological inferences. Secondly, online assessments (which was the method used to collect data during the lockdown period) inherently carry bias and are less reliable. However, similar longitudinal studies which would further delineate the real effect of the COVID-19 pandemic on burnout and mental health of the medical students are lacking. Finally, the analysis was not adjusted for other factors, such as family or interpersonal stress and/or personality traits that could have possibly confounded the relationship between lockdown and the subsequent digital learning with burnout and mental health.

## 7. Directions for the Future

During this unparalleled pandemic, new education techniques developed in recent years have been widely used which might trigger new ways of education in the years to come. However, each student, each educator, and each training program are different, and therefore, the advantages of disadvantages of permanently adopting such new ways of education should be appropriately weighted. Medical studies differ significantly from studies in other fields not only because of their total duration, but mainly because it is necessary for medical students to be trained in a real clinical setting.

Our findings are useful, particularly for training program directors, as they make it clear that at least for medical students in their last year, medical training should not be virtual. Moreover, our study showed that medical students in their first year of clinical training (in our school, year 4 of medical studies) are significantly stressed and were relieved when clinical training was discontinued. This suggests that the training program could be revised. Reduced hours in the clinical setting in the first clinical year might have a positive effect on reducing the stressors for those students. Moreover, all medical schools should have easily accessible medical student mental health services. Some schools of medicine provide such services through departments of psychiatry or other associated training programs (i.e., psychology). Since this stressful lifestyle often continues through residency training and life as a physician, this is a critical period in which to develop and utilize functional and effective coping strategies [28].

## 8. Conclusions

Our study shows that digital learning in medical studies might carry significant risks for students. We found that not only did mental health deteriorate, but also cynicism levels increased. Particularly final year students struggled more, possibly due to the lack of clinical exposure just before they were to start working as qualified junior doctors.

Preventing burnout is crucial. Educators and medical studies program directors should be aware that digital learning may have a detrimental effect on medical students’ burnout levels and overall mental health and weigh the risks when implementing such a way of medical education.

As it is, at the writing of this paper, unknown when the pandemic will be over, the long-term effects are yet to be determined. However, it is clear already that even if the current temporary measures come to be adopted as permanent new ways of education in some schools (i.e., complete digital learning), they should not in medical schools.

## Figures and Tables

**Figure 1 ijerph-18-00349-f001:**
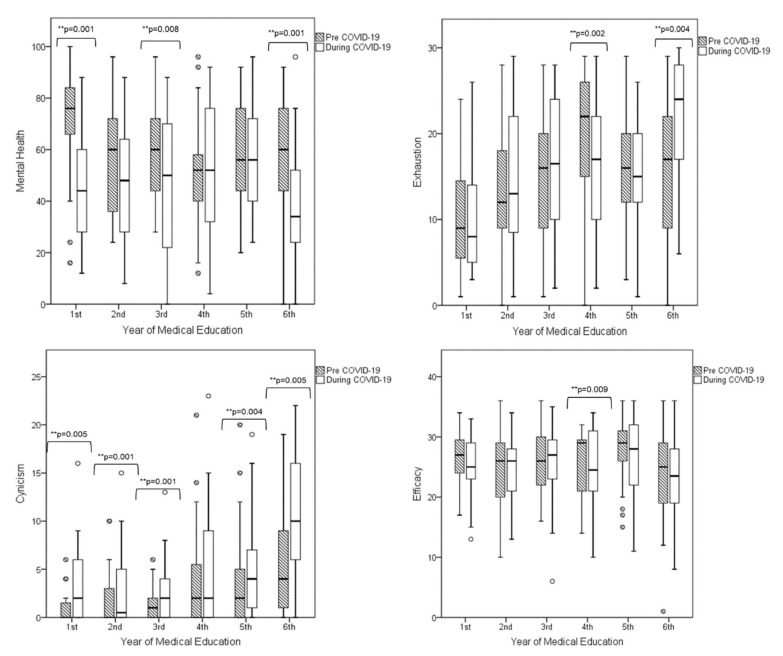
Changes observed in Maslach Burnout Inventory Student Survey dimensions and overall mental health after implementing digital learning.

**Table 1 ijerph-18-00349-t001:** Burnout prevalence across academic years in the different COVID-19 periods.

	Total	1st	2nd	3rd	4th	5th	6th
Students with Burnout during COVID-19 period	28/154 (18.2)	0/18 (0)	5/28 (17.9)	5/32 (15.6)	4/24 (16.7)	1/26 (3.8)	13/26 (50)
Students with Burnout during pre-COVID-19 period	33/182 (18.1)	1/27 (3.7)	6/33 (18.2)	3/37 (8.1)	11/27 (40.7)	4/29 (13.8)	8/29 (27.6)
Sig. ^1^	0.5	0.507	0.5	0.108	0.011 *	0.118	0.01 *

Burnout prevalence (n/N, %) among medical students in different COVID-19 periods and academic years. ^1^ Binomial tests. * *p* ≤ 0.02: Benjamini–Yekutieli corrected level of significance for six comparisons.

## Data Availability

All databases are available from the corresponding author upon reasonable request.

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
