# Peer review of "Medical Studies during the COVID-19 Pandemic: The Impact of Digital Learning on Medical Students’ Burnout and Mental Health"

_ijerph, 2021, doi:10.3390/ijerph18010349_

Round 1

Reviewer 1 Report

The aim of this ecological study (mini review) was to investigate what was the impact of digital learning which was implemented because of the COVID-19 pandemic on the burnout and the overall mental health of medical students at the University of Cyprus. This study needs several major revisions. In particular, authors should describe the design and audience of the study more specifically in the Methods' section.If the authors complete major revisions, the quality of the study will be further improved.

1. Title: It would be nice to include'medical students' in the title.

2. line 80-96: Authors should describe the study design more specifically. For example, Authors should describe about the sample's sampling. In addition, the authors must describe the non-response rate of the questionnaire in the'Method' section.

3. line 125-136: Authors should describe about the statistical tests. In particular, you should also explain why you chose the nonparametric test.

4. The control variable (or confound variable) used in this study should be specifically addressed. 

5. The conclusion ("Digital learning in medical studies carries significant risks.)  is too assertive. In the discussion section, the author should describe several possibilities for the subject's burnout. 

Author Response

  1. Title: It would be nice to include' medical students' in the title.

We have included “medical students’” in the title as requested.

  1. line 80-96: Authors should describe the study design more specifically. For example, Authors should describe about the sample's sampling. In addition, the authors must describe the non-response rate of the questionnaire in the'Method' section.

In the revised text we defined the study as ecological, we described the  response rate in the methods as requested.

  1. line 125-136: Authors should describe about the statistical tests. In particular, you should also explain why you chose the nonparametric test.

Non-parametric tests were used due to violation of the normality assumption. This was clarified in the revised section of “statistical analyses”

  1. The control variable (or confound variable) used in this study should be specifically addressed. 

We didn’t use any control variable. The “adjusted standardized residual” is a metric used during the crosstabs procedure in tables with larger than 2X2 design, in order to detect significant differences between actual and expected frequencies. Univariate analysis tests were solely performed in this study. This was clarified more in the revised text.

  1. The conclusion ("Digital learning in medical studies carries significant risks.)  is too assertive. In the discussion section, the author should describe several possibilities for the subject's burnout. 

We have changed the sentence which was quite assertive and we have added in the discussion, in the limitations section, other factors that may gave played a role.

Reviewer 2 Report

Dear authors, your manuscript is interesting but I need you to answer some questions:

INTRODUCTION

  • The introduction is very short. The constructs and concepts necessary to understand the manuscript are not explained. I suggested better explaining the dimensions of burnout.

METHODS

Study design:

  • The authors must specify the research design.
  • What was the target population? How was the sample chosen? The authors must specify it.

DISCUSSION

  • There are no references in paragraphs three to six (lines 203-235). Authors must correctly place bibliographic references.
  • The authors have not discussed the 'limitations of the study'. It is important to include them at the end of the "discussion".

REFERENCES

  • Many bibliographies are obsolete and some citations are incomplete. The bibliographic citations used are more than 5 years old (50%). The authors must update and arrange the bibliography.
  • There is an updated bibliography of original and meta-analytic articles that should be cited, among others (e.g. in the first reference).
  • The references do not meet the journal guidelines and some references that have errors. You should not put the month of publication or the PMID / PMCID. The authors should review this section.

Author Response

INTRODUCTION

  • The introduction is very short. The constructs and concepts necessary to understand the manuscript are not explained. I suggested better explaining the dimensions of burnout.

We have tried to keep introduction short and to the point. However, we have made some changes in order to explain better the basic concepts as suggested.

METHODS

  • The authors must specify the research design.
  • What was the target population? How was the sample chosen? The authors must specify it.

In the revised text we defined the study as ecological and we provided further details of the sampling method.

DISCUSSION

  • There are no references in paragraphs three to six (lines 203-235). Authors must correctly place bibliographic references.

The reason for this is that in those 4 paragraphs we are discussing the findings of our own study. We have added. However, a reference that was missing when we referred to MBI-SS and its use.

  • The authors have not discussed the 'limitations of the study'. It is important to include them at the end of the "discussion".

We have added a limitations section as requested.

REFERENCES

  • Many bibliographies are obsolete and some citations are incomplete. The bibliographic citations used are more than 5 years old (50%). The authors must update and arrange the bibliography.

Allow us to disagree. The date of publication does not necessarily reflects quality. We have made however some changes in the references based your suggestions below.

  • There is an updated bibliography of original and meta-analytic articles that should be cited, among others (e.g. in the first reference).

We have changed the citations were applicable. If you have a specific paper in mind that you feel would further improve our paper please let us know and we will include it

  • The references do not meet the journal guidelines and some references that have errors. You should not put the month of publication or the PMID / PMCID. The authors should review this section.

We have amended the references accordingly.

Reviewer 3 Report

The work presented is very interesting. It is worth exploring how the outbreak of the COVID pandemic and the measures taken in higher education have affected university students, especially those who are training in health science degrees.

However, there are some aspects to consider to improve the work presented:

- The authors use two different citation systems so it is not possible to follow the references they use in the text. This is something extremely important to correct.
- Page 2, line 60: "Current literature supports a strong link between burnout in medical students and increased 60 suicidality". However, only one reference is mentioned at the end of the paragraph. It is necessary to expand the bibliographic review and the literature on these aspects for a better understanding of the phenomenon.
- With respect to methodology, it is not known how the questionnaires administered to students were matched. More information on the procedure is needed.
- The results are well presented.
- It would be interesting to explain how the curriculum is organized in order to better assess not only the impact of the virtual methodology but also the workload of the students (and whether this increased during the time of training exclusively online).

Author Response

- The authors use two different citation systems so it is not possible to follow the references they use in the text. This is something extremely important to correct.

We have amended the references accordingly.

- Page 2, line 60: "Current literature supports a strong link between burnout in medical students and increased 60 suicidality". However, only one reference is mentioned at the end of the paragraph. It is necessary to expand the bibliographic review and the literature on these aspects for a better understanding of the phenomenon.

Accidentally we haven’t inserted the correct citation in that paragraph. We have done so and we added one more key reference to support the statements.

- With respect to methodology, it is not known how the questionnaires administered to students were matched. More information on the procedure is needed.

This was an ecological study, meaning that questionnaires were not matched. This was already mentioned in the statistical analysis and is now further clarified in the revised manuscript.

- It would be interesting to explain how the curriculum is organized in order to better assess not only the impact of the virtual methodology but also the workload of the students (and whether this increased during the time of training exclusively online).

All students in the Medical School receive lectures, laboratory training and clinical visits in the inpatient and outpatient departments, starting at 8am and finishing at around 4pm each day. During the lockdown, students spent less hours in education since clinical training was postponed. Thus, the workload was significantly cut down especially for students during the last three academic years wherein patient visits normally occupy about two thirds of their education. This could explain the decrease of burn-out rates among students in the 1st clinical year (i.e., 4th academic) during the COVID-19 period. On the other hand, as elaborated in the discussion part, this phenomenon was not observed among the 6th year students presumably due to the graduation-related stress. To address the comment, more information about the curriculum change due to the lockdown was added in the discussion part.

Round 2

Reviewer 1 Report

Thanks for recommending me as a reviewer. The aim of this ecological study (mini review) was to investigate what was the impact of digital learning which was implemented because of the COVID-19 pandemic on the burnout and the overall mental health of medical students at the University of Cyprus. The authors have completed the revision. This research deserves publication in IJERPH.

Reviewer 2 Report

Dear authors,

It´s true that the date is not a reflection of quality. However, there are many recent quality studies. If there are recent studies there is no reason to cite old papers.

Thanks for your reply. The explanations of the authors are satisfactory. The paper has greatly improved its quality.

Congratulations on your work.

Best regards

Reviewer 3 Report

The changes and comments made are suitable for publication of the manuscript in the journal.

This manuscript is a resubmission of an earlier submission. The following is a list of the peer review reports and author responses from that submission.